# Research on the Interaction Capability and Microscopic Interfacial Mechanism between Asphalt-Binder and Steel Slag Aggregate-Filler

Xiaobing Chen [1,2,*], Wei Wen [1,*], Jianguang Zhou [3], Xiaolong Zhou [4,*], Yunfeng Ning [3], Zhongshan Liang [5] and Zhenyu Ma [1]

1 School of Transportation, Southeast University, Nanjing 211189, China
2 Architects and Engineers Co., Ltd. of Southeast University, Nanjing 210096, China
3 Suzhou Jiaotou Construction Management Co., Ltd., Suzhou 215007, China
4 School of Civil Engineering and Architecture, Wuhan Institute of Technology, Wuhan 430073, China
5 Suzhou Sanchuang Pavement Engineering Co., Ltd., Suzhou 215124, China
* Correspondence: xbchen@seu.edu.cn (X.C.); 220203258@seu.edu.cn (W.W.); zhouxiaolong@wit.edu.cn (X.Z.)

**Abstract:** To explore the applicability of steel slag porous asphalt mixture, the interaction capability and microscopic interfacial mechanism between asphalt-binder and steel slag aggregate-filler were investigated in this laboratory study. These objectives were accomplished by comparing and analyzing the differences between steel slag and basalt aggregates in interacting with the asphalt-binder. The study methodology involved preparing basalt and steel slag asphalt mortar to evaluate the penetration, ductility, softening point, toughness, and tenacity. Thereafter, the interaction capability between the asphalt-binder and aggregates was characterized using the interaction parameters of the asphalt mortar obtained from dynamic shear rheometer (DSR) testing. For studying the functional groups and chemical bonding of the asphalt mortar, the Fourier Transform infrared (FTIR) spectrometer was used, whilst the interfacial bonding between the asphalt-binder and aggregates was analyzed using the scanning electron microscope (SEM). The corresponding test results indicated that the physical and rheological properties of the two asphalt mortars were similar. However, whilst the FTIR analysis indicated domination through chemical reactions, the interaction capability and interfacial bonding between the asphalt-binder and steel slag aggregates exhibited superiority over that between the asphalt-binder and basalt aggregates, with pronounced adsorption peaks appearing in the steel slag asphalt mortar spectrum. On the other hand, the SEM test revealed that, compared with the basalt, the micro-interfacial phases between the steel slag and asphalt-binder were more continuous and uniform, which could potentially enhance the interfacial bond strength between the asphalt-binder and aggregates (filler).

**Keywords:** asphalt-binder; steel slag; basalt; aggregate-filler: filler; interaction capability; microscopic interfacial mechanism; DSR; FTIR; SEM



## 1. Introduction

Asphalt mixture is a heterogeneous multiphase viscoelastic material consisting of asphalt-binder, fillers, and aggregates and has been widely used in the road construction industry due partially to its good constructability, performance, and durability [1,2]. In an asphalt mixture matrix system, fillers are dispersed in the asphalt-binder to form asphalt mortar, which plays a crucial role in the pavement performance of the asphalt mixture because the asphalt mortar serves to bond the aggregates, fill the voids, and transfer the loads [3]. However, the overall performance of an asphalt mixture predominantly depends on the properties and strength of the interfacial structure bonding between the asphalt-binder and aggregates [4]. On the other hand, the strength of these interfacial structure bonds partially depends on the asphalt-binder–filler interaction.

The asphalt-binder–filler interaction is a complex physical and chemical reaction that includes the physical adsorption between the asphalt-binder and filler; the chemical reaction on the contact surface of the asphalt-binder and filler; and the selective diffusion process of the asphalt-binder components in the asphalt mortar [5]. During the asphalt-binder–filler interaction, the chemical composition of the asphalt-binder rearranges itself around the filler surface. When the proportion of the asphalt-binder is small, the asphalt-binder can be completely converted into structural asphalt-binder that is adsorbed on the filler surfaces. When the asphalt-binder proportion is relatively high, however, the asphalt-binder outside this film thickness is called free asphalt-binder [6,7].

The volume ratio of the filler to the asphalt-binder is usually determined through the performance testing of the corresponding asphalt mixtures [8]. The physical and mechanical properties of the asphalt mortar reflect the asphalt-binder–filler interaction from a macroscopic perspective and can be readily analyzed using penetration, softening point, and ductility tests, among others [9,10].

The asphalt-binder–filler interaction has an important influence on the rheological properties of the asphalt mortar. The dynamic shear rheometer (DSR) and bending beam rheometer (BBR) are some of the commonly used test methods for studying the rheological properties of the asphalt mortar under high, intermediate, and low temperatures [11–16]. Rheological indices such as the rutting resistance parameter, fatigue cracking resistance parameter, and creep stiffness are typically used to analyze and quantify the influence of temperature, loading frequency, filler pH, filler content, and filler size on the asphalt-binder–filler interaction capability [17–19]. To explain the interaction mechanism between the asphalt-binder and filler, the composition and functional groups of the asphalt mortar are often analyzed using the Fourier transform infrared (FTIR) spectrometer [20,21]. Likewise, various microscopic testing techniques are available for studying the microstructure morphology of the asphalt-binder–filler interface and the surface morphology of the asphalt mortar. These techniques include fluorescence microscope (FM), scanning electron microscope (SEM), atomic force microscope (AFM), etc. [22–25].

The physical and chemical properties of filler are closely related to the properties of the asphalt mortar as well as the asphalt-binder–filler interaction. Alkaline filler has a stronger interaction with the asphalt-binder than the acidic filler [26]. For the same amount of asphalt-binder, the filler with a large specific surface area and more microporous structure generally has greater interaction with the asphalt-binder [27].

Limestone, fly ash, cement, etc. are usually used as fillers and admixtures in asphalt mixtures [28,29]. However, with the rapid development and worldwide construction of more transportation infrastructures, high-quality natural fillers are being exploited and consumed in larger quantities than ever before, leading to an urgent need for replacements. Steel slag filler is a high basicity filler, with strong adhesive bonding ability to asphalt-binder. Thus, as reported in the literature [30–32], steel slag utilization may valuably serve as an effective way to save the limited and diminishing natural resources. The chemical composition and mineral composition of steel slag are extremely complex, and the interaction between asphalt-binder and steel slag aggregate-filler depends on mechanical reinforcement and physicochemical interaction [33]. Moreover, the interaction between asphalt-binder and steel slag aggregate-filler is a very complex physicochemical interaction that occurs at the interface between asphalt-binder and steel slag aggregate-filler [34]. If the interface interaction between asphalt-binder and steel slag aggregate-filler is weak, the interface between asphalt-binder and steel slag aggregate-filler is easily damaged, thus leading to early pavement distress. Therefore, evaluating the interaction capability between asphalt-binder and steel slag aggregate-filler and understanding the interfacial mechanism between asphalt-binder and steel slag aggregate-filler are essential to improve the properties of asphalt mortar and the performance of asphalt mixture.

Whilst numerous studies have been conducted and documented on asphalt mortar, studies on the interaction between steel slag aggregates (filler) and asphalt-binder are still limited. Therefore, in this study, steel slag and basalt asphalt mortars were prepared

with a fixed volume ratio of filler to asphalt-binder, namely 40% (i.e., 0.4). Thereafter, the influence of steel slag and basalt fillers on the basic physical properties of the asphalt-binder were investigated using penetration, ductility, softening point, toughness, and tenacity tests. In the study, the rheological properties of two asphalt mortars were evaluated using the DSR test device, which indirectly aimed at determining and quantifying the interaction capability between the asphalt-binder and aggregates. For characterizing the microscopic mechanisms, functional groups, microstructure morphology, and interfacial bonding between the asphalt-binder and steel slag aggregates, the FTIR and SEM tests were used.

Based on the foregoing challenges, limitations, and literature gaps, the overall goal of this study was to investigate, quantify, and optimize the interfacial bonding mechanisms between the asphalt-binder and steel slag aggregates for potential use in steel slag porous asphalt mixture applications. The second technical objective of the study was to comparatively characterize the physical properties, rheological properties, chemical compositions, and functional groups of two asphalt mortars and morphological characteristics between the asphalt-binder and aggregates, namely the blend admixture of: (a) asphalt-binder and basalt aggregates and (b) asphalt-binder and steel slag aggregates. These objectives were accomplished through extensive laboratory testing that included the traditional asphalt-binder tests, DSR temperature sweep and time sweep tests, FTIR spectra analysis, and SEM microstructure imaging. Note that in this paper, the term "aggregates" refers to fine-ground aggregates passed through a 0.15 mm sieve and retained on the 0.075 mm sieve, called, respectively, filler or aggregate-filler.

## 2. Materials and Methods

### 2.1. Raw Materials

The styrene–butadiene–styrene (SBS) modified asphalt-binder used to prepare the asphalt mortars was sourced from Suzhou Sanchuang Road Engineering Co., Ltd., Suzhou, China. The content of the SBS modifier in the modified asphalt-bonder matrix was 4.5% by weight (i.e., 4.5 wt%) of the asphalt-binder. As listed in Table 1, the physical properties of the SBS modified asphalt-binder were measured following the Chinese standard JTG E20-2011 [35] and satisfactorily met the technical requirements of the Chinese specification JTG F40-2004 [36].

**Table 1.** Technical Indices of the SBS Modified Asphalt-Binder.

| | Index | Units | Test Results | Spec Requirement [36] |
|---|---|---|---|---|
| | Penetration (25 °C, 100 g, 5 s) | 0.1 mm | 55.9 | 40~60 |
| | Penetration index (PI) | - | 0.2 | −0.2~+1.0 |
| | Ductility (5 cm/min, 5 °C) | cm | 34.6 | ≥20 |
| | Softening point (ring and ball method) | °C | 82.5 | ≥70 |
| | Density | g/cm$^3$ | 1.029 | - |
| After TFOT [1] | Mass variation | % | 0.14 | ≤±1.0 |
| | Softening point difference (After-before) | °C | −4 | −12~+10 |
| | Penetration ratio (25 °C) | % | 80 | ≥65 |
| | Ductility (5 °C) | cm | 22.6 | ≥15 |

[1] Thin Film Oven Test.

Basalt and steel slag sourced from Hainan and Jiangsu Yonggang Group Companies (China), respectively, were selected as the aggregates for use in this study. The aggregate's physical properties were measured according to the Chinese standard JTG E42-2005 [37] and assessed for technical compliance using the Chinese specification JTG F40-2004 [36]. The corresponding technical indices are summarized in Table 2.

Table 2. Technical Indices of the Basalt and Steel Slag Coarse Aggregates.

| Index | Units | Basalt | Steel Slag | Spec Requirement [36] |
|---|---|---|---|---|
| Apparent specific gravity | - | 2.900 | 3.549 | $\geq$2.60 |
| Water absorption | % | 0.47 | 1.59 | $\leq$2.0 |
| Crush value | % | 10.4 | 13.4 | 13.4 |
| Los Angeles abrasion value | % | 14.6 | 10.7 | 10.7 |
| Flat elongated particles content | % | 9.8 | 10.1 | 10.1 |
| Water washing method (<0.075 mm particle Content) | % | 0.47 | 0.47 | 0.47 |
| Adhesion | - | 5 | 5 | 5 |
| Polishing value | - | 49 | 52 | 52 |

*2.2. Experimental Test Methods*

2.2.1. Preparation of the Asphalt Mortar

The volume ratio of filler to asphalt-binder (i.e., F/A) greatly affects the asphalt-binder-filler interaction ability. From a literature report [38], the recommended F/A ratio for obtaining the optimal asphalt-binder–filler interaction effects was 0.4~0.5 [38]. The F/A ratio used in this study was 0.4. Note, however, that being outside these limits (i.e., approaching the critical volume fraction) has the potential to negatively affect the thickness of the asphalt-binder film around the filler surface [39]. The preparation steps for both the basalt asphalt mortar (i.e., basalt mortar) and steel slag asphalt mortar (i.e., steel slag mortar) were as follows:

Step 1: After washing and drying, 5000 g each of steel slag and basalt coarse aggregates were ground into filler using an electromagnetic sample pulverizer (Wangsheng Instrument Factory, Shaoxing, China). Thereafter, the fillers (namely the finely grinded aggregates) passing through the 0.15 mm sieve and retained on 0.075 mm sieve (about 2500 g) were sampled for subsequent use in the production of the mortars.

Step 2: The asphalt-binder was heated in the oven at 170 °C for 2 h and then divided into various samples of about 1500 g. A thermostatic electric heating sleeve was used for heating and constantly maintaining the sample temperature at 170 °C.

Step 3: A specific weighed amount of fillers corresponding to the F/A ratio of 0.4 was, thereafter, added into the hot asphalt-binder gradually at a low rotational speed of 1000 rpm for 20 min. The blend mixtures were then continuously blended at 4000 rpm for about 30 min after all the fillers were added to obtain a homogeneously distributed asphalt mortar matrix.

Step 4: Lastly, the prepared asphalt mortars at 170 °C were poured into standard experimental molds for subsequent laboratory testing.

2.2.2. Physical Property Testing

To evaluate the influence of different fillers on the asphalt mortar, the physical properties of two asphalt mortars (namely basalt mortar and steel slag mortar) were tested and measured according to the Chinese standard JTG E20-2011 [35]. These physical tests included penetration, ductility, softening point, toughness, and tenacity. For each test, three sample replicates were tested per asphalt mortar type [40].

2.2.3. Rheological Property Testing

To evaluate the rheological properties of the asphalt mortars, the high and intermediate temperature sweep tests were conducted in strain-controlled loading mode at a constant frequency of 10 rad·s$^{-1}$ using a Malvern Kinexus (UK) DSR test device [16,41,42]. For the high-temperature sweep tests, a parallel plate with 25 mm diameter and 1 mm gap was used. The temperature range was 88~118 °C with an incremental interval of 6 °C at a controlled strain rate of 12%. For the intermediate temperature sweep tests, a parallel plate with 8 mm diameter and 2 mm gap was used. The temperature range was 16~31 °C with

an incremental interval of 3 °C at a controlled strain rate of 1%. Thereafter, all DSR test data, with three sample replicates per test condition per asphalt mortar type, were measured, processed, and analyzed using rSpace software [43,44].

### 2.2.4. Fourier Transform Infrared (FTIR) Spectrometer Testing

The chemical composition of asphalt-binder, aggregate filler, and asphalt mortar were determined using a Nicolet IS10 fourier transform infrared (FTIR) spectrometer (Thermo Fisher Scientific, Waltham, MA, USA). During FTIR testing, the potassium bromide (KBr) pressing plate method was used for the filler, whilst a solution method was utilized for the SBS-modified asphalt-binder and the asphalt mortar [42,45]. The spectra data were measured over a wavenumber range of 400 $cm^{-1}$ to 4000 $cm^{-1}$, with a resolution of 2 $cm^{-1}$. All material spectra data, with three test replicates per test condition per asphalt mortar type, were collected and processed using the OMNIC software [46,47].

### 2.2.5. Scanning Electron Microscope (SEM) Testing

The morphological macrotexture of the aggregate surface and the microstructure of the asphalt-binder-aggregate interface transition zone (ITZ) were analyzed and quantified using the scanning electron microscope (SEM, FEI Company, Eindhoven, The Netherlands). In the study, all samples were successively sputter coated with a thin gold film prior SEM imaging analysis [23,48]. Three surface locations of each sample were scanned with various magnifications of ×200, ×500 and ×1000, then a representative surface location of each sample was selected for the analysis [3].

## 3. Laboratory Test Results and Analysis

### 3.1. Physical Properties of the Asphalt Mortar

Penetration and softening point reflecting the consistency and the constant temperature viscosity were used to evaluate the high-temperature deformation resistance of the asphalt mortar. Ductility was used to evaluate the low-temperature plastic deformation ability of the asphalt mortar [49,50]. The corresponding test results of these physical properties are presented in Figure 1.

After replacing an equal volume of the basalt fillers with steel slag fillers, the penetration and ductility of the asphalt mortar decreased by 8.8% and 12%, respectively, whilst the softening point increased by 3.1%. By comparison, Figure 1 shows that the high-temperature deformation resistance of the steel slag asphalt mortar was better than that of the basalt asphalt mortar and vice versa for low-temperature plastic deformation ability. Considering that the absolute values of the physical indices for the asphalt mortars were small, the resultant differences in the physical properties of two asphalt mortars were observed to be equally marginal and insignificant. Compared with the basalt asphalt mortar, the toughness of steel slag asphalt mortar increased by 10.1%, whilst the tenacity decreased by 27.7%. This demonstrated that the steel slag asphalt mortar had better adhesion capability with the aggregates (filler) than the basalt asphalt mortar.

### 3.2. Rheological Properties of the Asphalt Mortar

The accumulation of compressive and shear strains in the asphalt mixture is considered one of the major sources of asphalt-mixture layer rutting in flexible asphalt pavements and is often simulated as a stress-controlled cyclic loading phenomenon [4,51,52]. During the deformation of the asphalt mixture and due to its viscoelastic nature (i.e., the asphalt-binder in it), the work expanded by traffic loading is partially recovered by the elastic component of the strain and partially dissipated by the viscous flow component of the strain [53,54]. On the other hand, the fatigue cracking of flexible asphalt pavements at intermediate temperatures could be either a stress- or strain-controlled phenomenon in thick or thin asphalt-mixture surfacing layers, respectively [55–57]. So, for thin asphalt-mixture surfacing layers, the potential occurrence of this distress mode (i.e., fatigue cracking) is

predominantly attributed to the strain-controlled loading phenomenon and vice versa for thick asphalt-mixture surfacing layers.

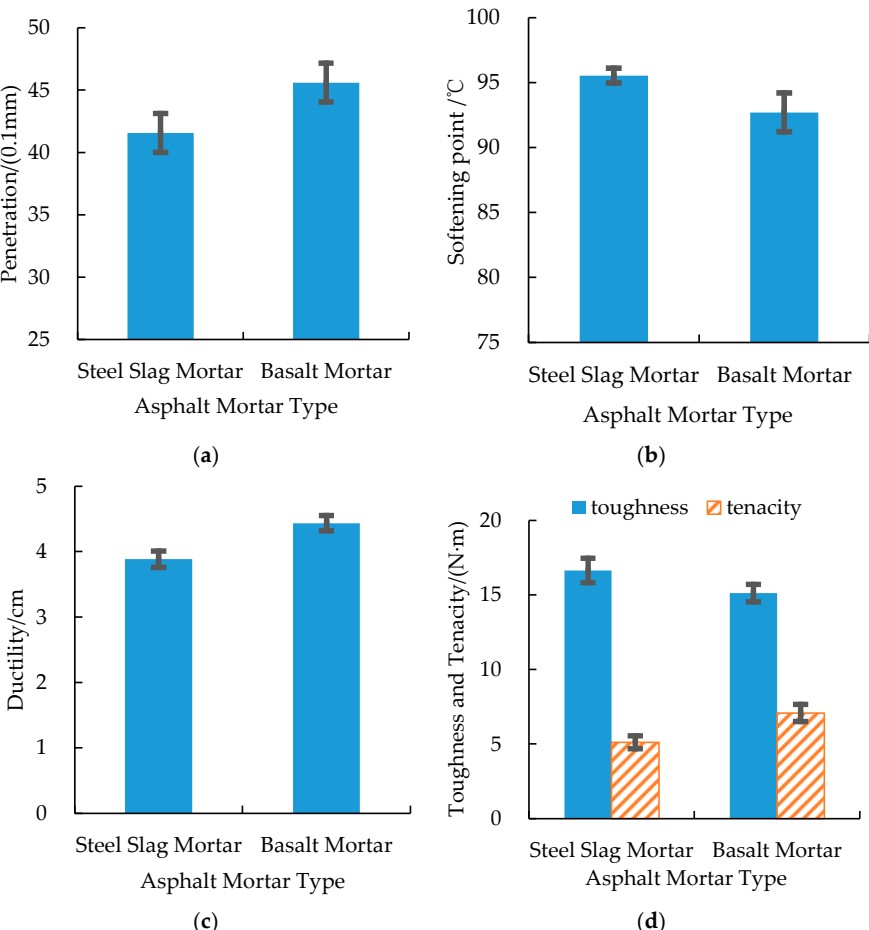

**Figure 1.** Physical properties of the asphalt mortar: (**a**) Penetration; (**b**) Softening Point; (**c**) Ductility; (**d**) Toughness and Tenacity.

Based on the dissipated energy concept and the controlled stress hypothesis, Anderson [58] suggested using the rutting resistance parameter ($G^*/\sin\delta$) for evaluating the high-temperature rheological properties and rutting resistance potential of viscoelastic materials such as asphalt-binder. Likewise, a fatigue cracking resistance parameter ($G^*\cdot\sin\delta$), based on the dissipated energy concept and the controlled strain hypothesis, was formulated to evaluate the intermediate-temperature rheological properties of asphalt-binder. The mathematical models for computing these rheological parameters are expressed in Equations (1) and (2), respectively [58]:

$$W_c = \frac{\pi \cdot \sigma_0^2}{G^*/\sin\delta},\tag{1}$$

$$W_c = \pi \cdot \varepsilon_0^2 \cdot (G^* \cdot \sin\delta),\tag{2}$$

In Equations (1) and (2), $W_c$ = the work dissipated per load cycle, $\sigma_0$ = the stress applied during the load cycle, $\varepsilon_0$ = the strain during the load cycle, $G^*$ = the complex shear modulus, and $\delta$ = the phase angle. Equation (1) mathematically indicates that to minimize rutting deformation, $G^*/\sin\delta$ should be increased. Similarly, Equation (2) shows that a quantitative decline in $G^*\sin\delta$ will potentially minimize fatigue cracking [58].

Mathematically, Equation (1) indicates that the work dissipated per loading cycle is inversely proportional to the rutting resistance parameter. Thus, the larger the rutting resistance parameter is in magnitude, the greater the rutting resistance the asphalt-binder (or asphalt

mortar) is at high temperatures and vice versa [51]. The corresponding DSR test results for the high-temperature rheological properties of two asphalt mortars are shown in Figure 2.

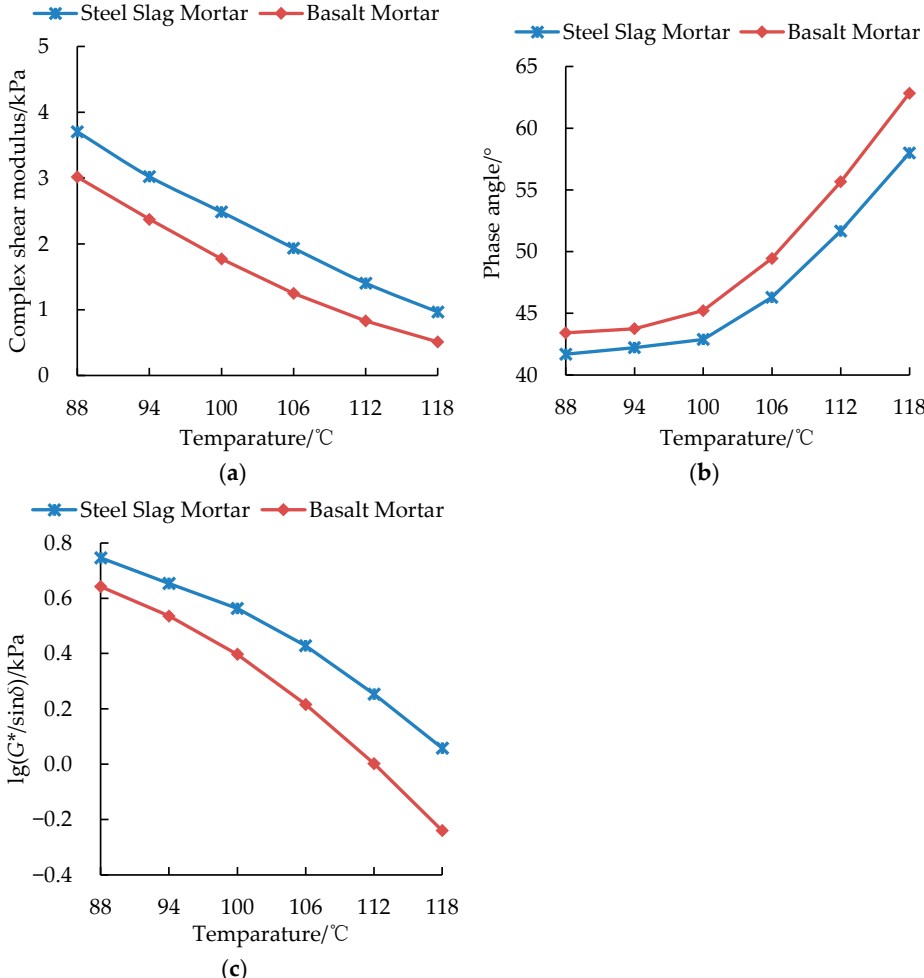

**Figure 2.** High-temperature rheological property results: (**a**) Complex shear modulus; (**b**) Phase angle; (**c**) Rutting resistance parameter.

As shown in Figure 2, when the DSR test temperature was increased, the complex shear modulus of the asphalt mortars decreased, with an exponential decline in the rutting resistance parameter ($G^*/\sin \delta$)—indicating a decay in deformation resistance. As a theoretically expected response behavior of viscoelastic materials, the phase angle correspondingly increased with an increase in temperature. Compared with the basalt asphalt mortar, the steel slag asphalt mortar had a smaller phase angle, larger complex shear modulus, and larger rutting resistance parameter at the same DSR test temperature. At high temperature, the steel slag asphalt mortar exhibited greater stiffness with lower fluidity, indicating that the asphalt-binder and steel slag fine aggregates had superior interaction than that between the asphalt-binder and basalt fine aggregates.

By and large, the differential value between the rutting resistance parameters of the two asphalt mortars increased continuously as the DSR test temperature was increased. This suggested that the rutting resistance parameter of the basalt asphalt mortar was more sensitive to temperature. Therefore, the high-temperature rheological properties of the steel slag asphalt mortar with respect to rutting resistance potential were deemed superior to that of the basalt asphalt mortar.

The work during a loading cycle could be dissipated in one or more of the following damage mechanisms, namely cracking, crack propagation, heat dissipating, plastic flow, etc. [58]. As previously seen in Equation (2), the work dissipated per loading cycle is numerically

proportional to the fatigue cracking resistance. Therefore, the smaller the fatigue cracking resistance is in magnitude, the greater the fatigue cracking resistance the asphalt binder (or asphalt mortar) is at intermediate temperatures and vice versa [59]. The corresponding DSR test results for the intermediate-temperature rheological properties of two asphalt mortars are shown in Figure 3.

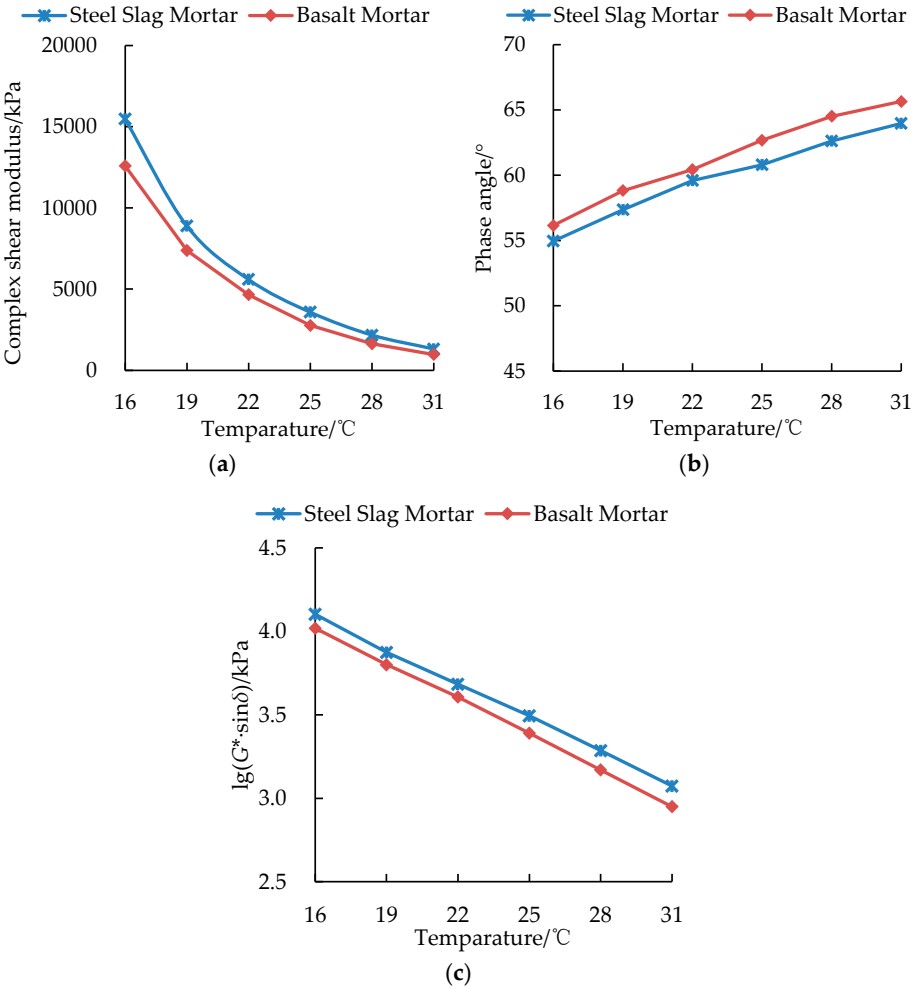

**Figure 3.** Intermediate-temperature rheological property results: (**a**) complex shear modulus; (**b**) phase angle; (**c**) fatigue crack resistance parameter.

As shown in Figure 3, when the DSR test temperature was increased, the complex shear modulus decreased, the phase angle increased, and the fatigue cracking resistance parameter ($G^*\cdot\sin\delta$), on a semi-log scale, decreased linearly. Theoretically, the higher the temperature is, the larger the volume of the free asphalt-binder is, with a corresponding smaller elastic component ratio for the asphalt mortar and larger viscous component ratio for the asphalt mortar [60,61]. Compared with the basalt asphalt mortar, the steel slag asphalt mortar had slightly smaller phase angles, slightly larger complex shear modulus, and slightly greater fatigue cracking resistance at the same DSR test temperatures. However, compared with Figure 3 for the high-temperature results, the differences in the intermediate-temperature rheological properties between steel slag and basalt asphalt mortar were relatively small but nonetheless indicated basalt mortar superiority over steel slag mortar with respect to fatigue cracking resistance potential.

### 3.3. Asphalt-Binder-Filler Interaction Capability

3.3.1. Parametric Index Formulation

The asphalt-binder–filler interaction, that is partially caused by the materials' non-Newtonian fluid characteristics at high temperatures [62–64], can potentially affect the rheological properties and ultimate performance of the asphalt mortar. To evaluate the asphalt-binder–filler interaction capability, numerous quantitative indices based on the rheological characterization were explored.

Usually, when the interactions are strong, the flow capacity of the asphalt mortar becomes weak, with its complex shear modulus $G*$ and complex viscosity $\eta*$ increasing in magnitude—and vice versa for weak interactions [65]. Therefore, the $G*$ and $\eta*$ rheological parameters were assumed to potentially reflect and indirectly be used as quantitative indicators of the asphalt-binder–filler interactions. With this assumption, the analysis in this study was theorized on the consideration that the greater $G*$ and $\eta*$, the stronger the interaction capability between the asphalt-binder and aggregate. To discount the effects of the asphalt-binder on the rheological properties of asphalt mortar, $G*$ and $\eta*$ were normalized to the complex shear modulus coefficient $\Delta G*$ and complex viscosity coefficient $\Delta\eta*$, respectively, as illustrated in Equations (3) and (4) [66]:

$$\Delta G^* = (G_m^* - G_b^*)/G_b^*, \tag{3}$$

$$\Delta \eta^* = (\eta_m^* - \eta_b^*)/\eta_b^*, \tag{4}$$

In Equation (3), $\Delta G*$ = the complex shear modulus coefficient, $G_m^*$ = the complex shear modulus of asphalt mortar (kPa), and $G_b^*$ = the complex shear modulus of asphalt-binder (kPa). In Equation (4), $\Delta\eta*$ = the complex viscosity coefficient, $\eta_m^*$ = the complex viscosity of asphalt mortar (kPa), and $\eta_b^*$ = the complex viscosity of asphalt-binder (kPa).

Ibrarra [67] proposed that the parameter Luis Ibrarra-$A$-$\delta$ ($L$-$A$-$\delta$) could be used to analyze the interfacial energy loss of composite materials. Based on a three-phase model characterization [67], the loss factor $\delta_c$ of composite materials could be approximated using Equation (5):

$$\tan \delta_c = \varphi_f \tan \delta_f + \varphi_i \tan \delta_i + \varphi_x \tan \delta_x \tag{5}$$

In Equation (5), $\delta_c$ = the phase angle of composite materials (°); $\delta_f$ = the phase angle of filling phase (°); $\delta_i$ = the phase angle of interfacial phase (°); $\delta_x$ = the phase angle of matrix phase (°); and parameters $\varphi_f$, $\varphi_i$, and $\varphi_x$ = the volume fractions of the filling phase, interfacial phase, and matrix phase, respectively (%).

Although Equation (5) cannot provide a detailed prediction of the loss factor in composite materials, it can potentially be used to compare the influence of the fillers and matrix interface with different treatments on the adhesion effect. In this regard, the $\tan \delta_f$ parameter would be assumed to be zero and the volume fraction of the interfacial phase negligible [7,66,68]. With these assumptions, Equation (5) can be simplified and reduced to Equation (6) as follows:

$$\tan \delta_c / \tan \delta_x = (1 - \varphi_f)(1 + A), \tag{6}$$

where $A$ = $L$-$A$-$\delta$, namely the filler-matrix interaction parameter. For asphalt mortar, parameter A could be expressed as shown in Equation (7) [66]:

$$L - A - \delta = \frac{\tan \delta_m}{\tan \delta_b (1 - \varphi)} - 1, \tag{7}$$

where $\delta_m$ = the phase angle of the asphalt mortar (°) and $\delta_b$ = the phase angle of asphalt-binder (°). Theoretically, a smaller value of $L$-$A$-$\delta$ is indicative of better bonding effects between the filler and the asphalt-binder, good interaction capability between the asphalt-binder and aggregates, and ultimately, greater constraint for the molecular movement near the interface. For an ideal interfacial bonding, the $L$-$A$-$\delta$ parameter should be practically zero (0), i.e., $L$-$A$-$\delta$ = 0 [67].

Ziegel [69] suggested the K.Ziegel-*B*-δ (*K-B-δ*) parameter to estimate the loss factor of a two-phase filling system using the model expressed in Equation (8):

$$\tan \delta_{\text{c}} = \tan \delta_{\text{x}} / (1 + \varphi_{\text{f}} B), \tag{8}$$

where *B* = *K-B-δ*. For the asphalt mortar, parameter *B* can be expressed as exemplified in Equation (9) [69]:

$$K - B - \delta = (\tan \delta_{\text{b}} / \tan \delta_{\text{m}} - 1) / \varphi, \tag{9}$$

However, the rheological characteristics of multiphase blended admixtures (or asphalt mortars) cannot be taken as a simple superposition of each phase as inferred by Equations (7) and (9) [70]. They are a function of and dependent on many other interactive factors such as the component concentration, interfacial tension, microdomain structure, and particle size and distribution. On this basis, it was therefore considered, in this study, that the *L-A-δ* and *K-B-δ* models could not adequately distinguish the asphalt-binder–filler interaction and the inter-particle interaction of the filler [70]. Therefore, the *K-B-G\** model was proposed to evaluate the asphalt-binder-filler interaction capability in this study [67,69].

The Palierne emulsion model [71] was one of the most commonly used models for quantifying the viscoelastic behavior of incompatible polymer blend materials. The model was formulated on the following two fundamental hypothesis:

- The dispersed phase presented a spherical distribution when in a continuous matrix phase.
- The interfacial tension between the dispersed phase and matrix phase had no relationship with the partial area change.

With the above hypotheses, the complex shear modulus of a blended matrix system (admixture or asphalt mortar) can be seen as a composite function of each phase complex shear modulus, dispersed phase particle size, and interfacial tension [71]. Furthermore, the particle size distribution in the dispersed phase matrix can be potentially replaced and substituted with the average particle size. At a fixed frequency, the Palierne emulsion model [71] can thus be expressed as shown in Equation (10):

$$G_{\text{c}}^* = G_{\text{x}}^* \frac{1 + 3\varphi H_{\text{d}}}{1 - 2\varphi H_{\text{d}}}, \tag{10}$$

In Equation (10), $G_{\text{c}}^*$ = the complex shear modulus of the composite materials, $G_{\text{x}}^*$ = the complex shear modulus of the matrix phase, $\varphi$ = the volume fraction of dispersed phase, $H_{\text{d}}$ = intermediate transition function, and $H_{\text{d}}$ can be determined as mathematically illustrated in Equation (11):

$$H_{\text{d}} = \frac{4(\alpha/R)\left(2G_{\text{x}}^* + 5G_{\text{d}}^*\right) + \left(G_{\text{d}}^* - G_{\text{x}}^*\right)\left(16G_{\text{x}}^* + 19G_{\text{d}}^*\right)}{40(\alpha/R)\left(G_{\text{d}}^* + G_{\text{x}}^*\right) + \left(2G_{\text{d}}^* + 3G_{\text{x}}^*\right)\left(16G_{\text{x}}^* + 19G_{\text{d}}^*\right)}, \tag{11}$$

where $G_{\text{d}}^*$ = the complex shear modulus of the dispersed phase, $\alpha$ = the interfacial tension, and *R* = the average particle size of the dispersed phase. For a hard particle matrix system that is dispersed in a viscoelastic substrate ($G_{\text{d}}^* \to \infty$, $H_{\text{d}} = 0.5$), the Palierne model can be simplified as expressed in Equation (12) [71]:

$$G_{\text{c}}^* = G_{\text{x}}^* \frac{1 + 1.5\varphi}{1 - \varphi}, \tag{12}$$

Ziegel [69] demonstrated that the volume fraction $\varphi$ could be replaced with $\varphi B$ when the dissipation energy was considered. In this regard, *B* represented the physicochemical

interaction between the dispersed and un-dispersed matrix phases. For the asphalt mortar applications in this study, Equation (12) was modified as follows:

$$K - B - G^* = \frac{G_m^*/G_b^* - 1}{\left(1.5 + G_m^*/G_b^*\right) \cdot \varphi}, \tag{13}$$

As with the rheological property evaluation in Section 3.2 of this paper, DSR time sweep testing of the steel slag and basalt asphalt mortars were conducted in strain-controlled loading mode using a Malvern Kinexus (UK) DSR test device. A parallel plate configuration, 25 mm in diameter and 1 mm gap, was used. The test was conducted at a constant temperature of 45 °C with a loading frequency of 10 rad·s$^{-1}$ and a controlled strain rate of 10% [16,42,72]. Thereafter, the interaction capability between the asphalt-binder and aggregate (steel slag or basalt) was quantitatively evaluated using the above indices, namely Equation (3) through Equation (13), i.e., $G_m^*$, $\Delta G^*$, $\eta_m^*$, $\Delta\eta^*$, $\delta$, *L-A-δ*, *K-B-δ*, and *K-B-G\**.

3.3.2. Parametric Index Results and Evaluation

As shown in Figure 4, the interaction parameters between the asphalt-binder and aggregates presents different response trends. Compared with the steel slag asphalt mortar, the complex shear modulus $G_m^*$, complex viscosity $\eta_m^*$, complex shear modulus coefficient $\Delta G^*$, complex viscosity coefficient $\Delta\eta^*$, *K-B-δ* and *K-B-G\** parametric indices of the basalt asphalt mortar were quantitatively smaller in magnitude, but vice versa for phase angle $\delta$ and *L-A-δ*. This suggested superior interaction capability within the steel slag mortar, namely asphalt-binder plus steel slag aggregate.

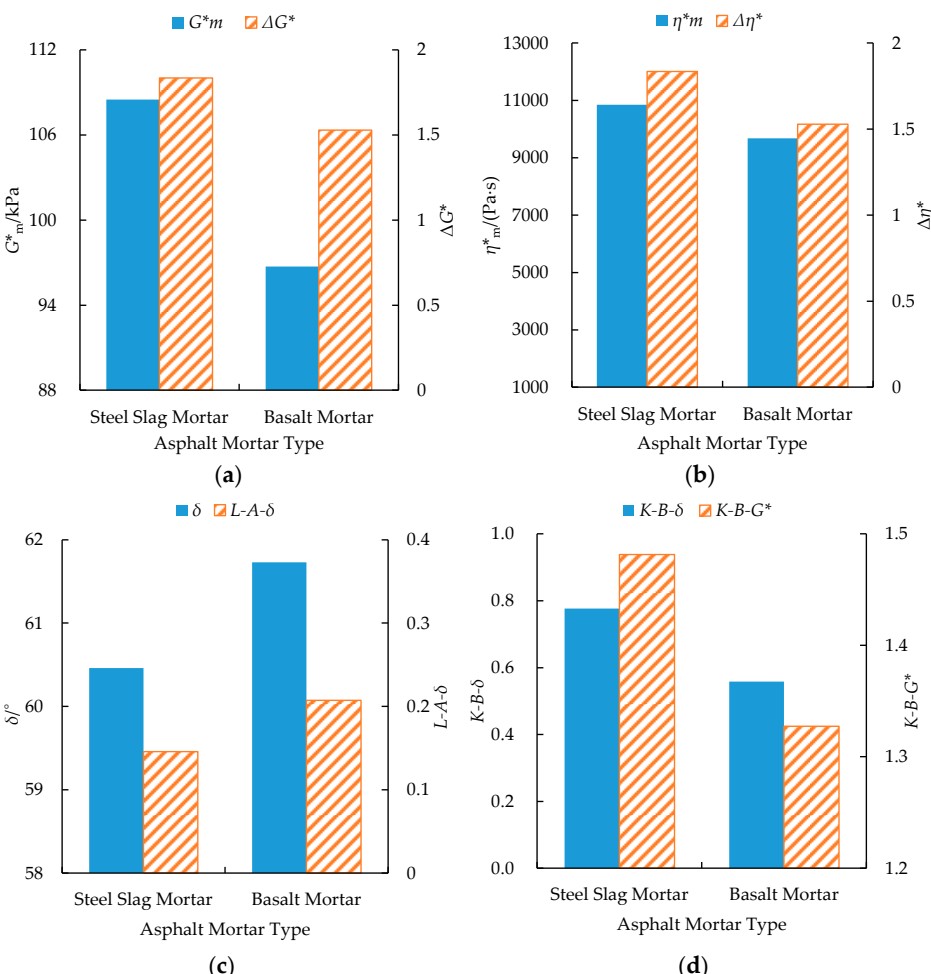

**Figure 4.** Index evaluation of the asphalt mortar interaction capability: (**a**) $G_m^*$ and $\Delta G^*$ indices; (**b**) $\eta_m^*$ and $\Delta \eta^*$ indices; (**c**) $\delta$ and $L$-$A$-$\delta$ indices; (**d**) $K$-$B$-$\delta$ and $K$-$B$-$G^*$ indices.

In general, the response trends and magnitudes of the parametric indices in Figure 4 demonstrated that the interaction capability between the asphalt-binder and steel slag aggregates was better than that between the asphalt-binder and basalt. Chemical reaction theory could be advanced to explain the influence of the acid–base property of the aggregates on asphalt-binder–aggregate interaction [73]. That is, the interaction between the asphalt-binder and aggregate filler was essentially a chemical reaction that resulted from the outermost orbital electrons of some active substances on the aggregate surface migrating to the active functional groups in the asphalt-binder. These active substances were rich in alkali aggregates but less in acidic aggregates, whilst the active substances in the neutral aggregates were in between with a pH around seven [74,75]. The major mineral components of steel slag were calcium hydroxide, RO phase dicalcium silicate, and tricalcium silicate, which made the steel slag more alkaline than basalt and able to react with the acid anhydride in asphalt-binder [76]. Therefore, compared with the basalt aggregates, the chemical reactions between the steel slag aggregates and the asphalt-binder were stronger as well as the corresponding interactions, which explains the superior interaction ability associated with the steel slag mortar.

### 3.4. Chemical Bonding between the Asphalt-Binder and Aggregate

The infrared spectra of the asphalt-binder, aggregates, and asphalt mortars were qualitatively analyzed using FTIR. The corresponding results are graphed in Figures 5 and 6 for steel slag and basalt, respectively.

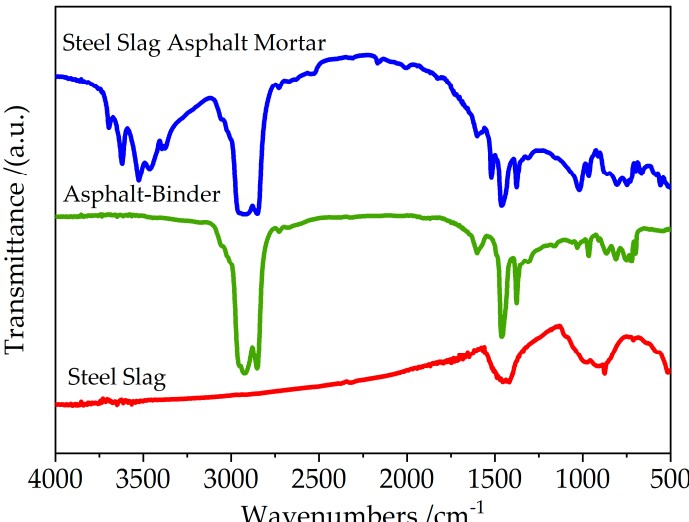

**Figure 5.** Infrared spectra of the asphalt-binder, steel slag, and mortar.

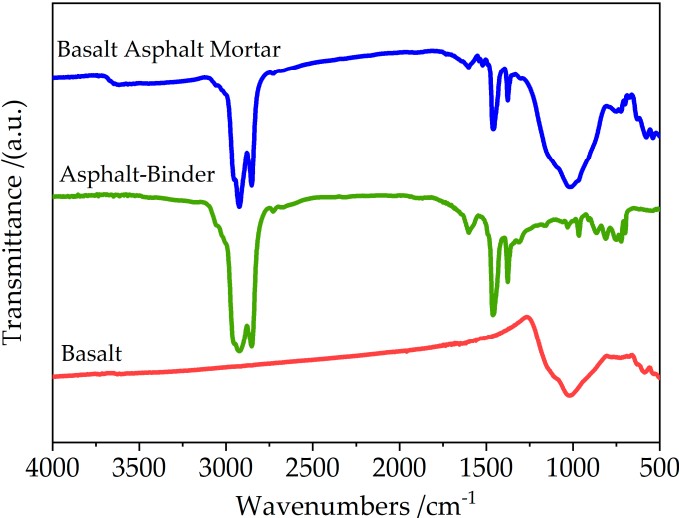

**Figure 6.** Infrared spectra of the asphalt-binder, basalt, and mortar.

Figure 5 shows that in the infrared spectrum functional group region ($4000 \sim 1330$ cm$^{-1}$) of the asphalt-binder, the strongest, moderate, and broad/weak bands were observed at 2923 cm$^{-1}$, 1461 cm$^{-1}$, and 1601 cm$^{-1}$, respectively, with the shoulder peaks of the strongest and moderate bands occurring at 2852 cm$^{-1}$ and 1376 cm$^{-1}$, respectively. The peaks at 2923 cm$^{-1}$ and 2850 cm$^{-1}$ were caused by the antisymmetric and symmetric stretching vibration of $CH_2$ in the alkanes, whilst the peaks at 1455 cm$^{-1}$ and 1376 cm$^{-1}$ were caused by the antisymmetric and symmetric variable angle vibration of $CH_3$ in the alkanes. The peaks at 1601 cm$^{-1}$ indicated the occurrence of the stretching vibration of C=C in asymmetric substituted aromatic ring.

As for the fingerprint region ($1330 \sim 500$ cm$^{-1}$) of the asphalt-binder, there existed some characteristic peaks at 1031 cm$^{-1}$, 966 cm$^{-1}$, 864 cm$^{-1}$, 811 cm$^{-1}$, 749 cm$^{-1}$, 722 cm$^{-1}$, and 699 cm$^{-1}$. The stretching vibration of sulfoxide S=O functional group appeared at 1031 cm$^{-1}$. The peaks at 966 cm$^{-1}$ and 699 cm$^{-1}$ were due to the out-plane bending vibration of the trans-CH and cis-CH in the alkenes whilst the peak at 722 cm$^{-1}$ was attributed to the in-plane sway vibration of $CH_2$ in the long-chain alkanes. On the other hand, the peaks at 864 cm$^{-1}$, 811 cm$^{-1}$, and 749 cm$^{-1}$ were caused by the out-plane bending vibration of CH in the benzene ring. The characteristic peaks at 966 cm$^{-1}$ and 699 cm$^{-1}$ that occurred in the asphalt-binder were attributed to the bending vibration of the C=C chains in the butadiene block of the SBS modifier. This demonstrated that the asphalt-binder contained alkane, cycloalkane, carboxylic acid, ester, amide, aliphatic amine,

aromatic ether, sulfoxide, and other functional compounds. The hydrophilic functional groups (such as carboxyl) enhanced the asphalt-binder to have the potential ability to combine and transport water, thereby weaking the chemical adsorption at the interface between the asphalt-binder and aggregate [77].

The broad and weak band of the steel slag was due to the stretching vibration of N-H or O-H bonds in the 3750~3400 $cm^{-1}$ wavenumber region. The absorption peak at 1420 $cm^{-1}$ was caused by antisymmetric stretching vibration in the carbonates, whilst the absorption peaks at 985 $cm^{-1}$, 910 $cm^{-1}$, 878 $cm^{-1}$, and 517 $cm^{-1}$ were attributed to the genetic map of inorganic compound, namely β-dicalcium silicate or tricalcium silicate containing $CaO$, $Al_2O_3$, $Fe_2O_3$. From these results and the infrared spectrum in Figure 5, it can be said that the inorganic compounds in the steel slag were iron oxide, calcium oxide, aluminum oxide, and silicon oxide.

Additionally, it can also be seen from the infrared spectrum of the steel slag asphalt mortar that most of the absorption peaks were essentially the superposition of the absorption peaks of the asphalt-binder and the steel slag aggregates, implying that physical action was the main action between the asphalt-binder and steel slag aggregates. Compared with the asphalt-binder, the absorption peaks at 2920 $cm^{-1}$ and 2852 $cm^{-1}$ caused by the stretching vibration of $-CH_2$ slightly decreased, whilst the absorption peak at 1601 $cm^{-1}$ was attributed to the slight decrease in the stretching vibration of the C=C chains in the aromatic ring, which indicated that the steel slag aggregates absorbed the light components of the asphalt-binder and increased the stiffness of the asphalt mortar.

There was a new broad and weak band between 3200 $cm^{-1}$ and 3750 $cm^{-1}$ that presumably was caused by the N–H stretching vibration of the amines and amides and the SiO–H stretching vibrations, respectively. Furthermore, there was also a new absorption peak at 1520 $cm^{-1}$ that was caused by the $-NO_2$ bond stretching vibration. This demonstrated that a chemical action existed between the asphalt-binder and the steel slag aggregates.

In general, the chemical bonding interaction was much stronger than the intermolecular force. This meant an enhancement in the bonding between the asphalt-binder and aggregate that ultimately contributed to the formation of a thicker asphalt-binder matrix with a significant improvement in the high-temperature rutting resistance and moisture resistance of the asphalt mortar. However, it can be seen from the infrared spectrum of the basalt asphalt mortar in Figure 6 that most of the absorption peaks were the superposition consequence of the absorption peaks of the asphalt-binder and basalt aggregates. The weak peak at 1520 $cm^{-1}$ was presumably caused by the $-NO_2$ bond stretching vibration, indicating that physical action was the main action between the asphalt-binder and basalt aggregates with a weak chemical adhesion occurring in the basalt asphalt mortar. From chemical reaction theory [73] and comparing with Figure 5, it can be explained that the adhesion between the steel slag aggregates and asphalt-binder was better than that between basalt aggregates and asphalt-binder. Thus, with the use of steel slag mortar, it can be theoretically assumed that this indication of good adhesive bonding will effectively prevent the loosening and peeling of the resultant asphalt mixture with an overall improvement in the moisture-damage resistance of the flexible asphalt pavement during its service life.

*3.5. SEM Imaging Results*

As can be seen in Figures 7 and 8, the steel slag had a rough surface texture and more pore structures, containing some elliptical or circular characteristic structures of basic oxygen furnace (BOF) slag. The basalt (Figure 8), on the other hand, had a dense and regular surface texture with few pore structures and smaller mineral crystals.

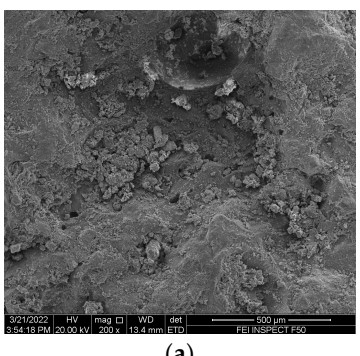

(**a**)

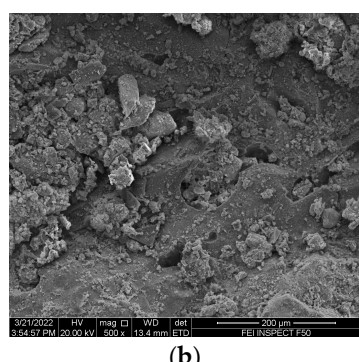

(**b**)

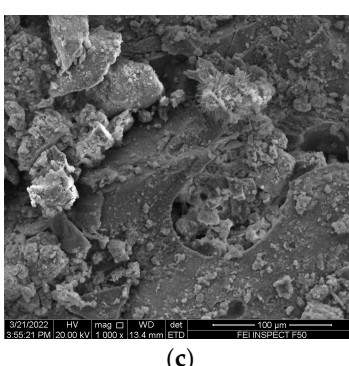

(**c**)

**Figure 7.** SEM surface morphology of steel slag aggregates: (**a**) ×200 Magnification; (**b**) ×500 Magnification; (**c**) ×1000 Magnification.

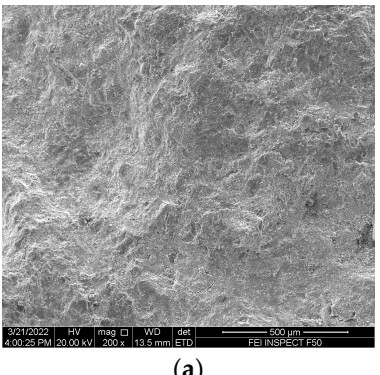

(**a**)

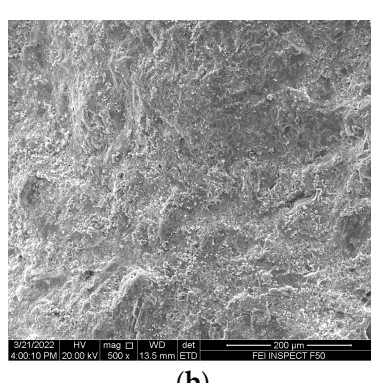

(**b**)

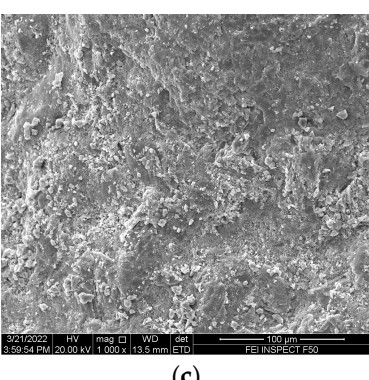

(**c**)

**Figure 8.** SEM surface morphology of basalt aggregates: (**a**) ×200 Magnification; (**b**) ×500 Magnification; (**c**) ×1000 Magnification.

In general, the SEM imaging results in Figure 7 indicate that the steel slag had a larger specific surface area and stronger asphalt-binder absorption capacity to form a more structurally strong asphalt mortar matrix than basalt. This meant that the steel slag could yield a relatively thicker asphalt-binder film. Furthermore, the characteristic pore structures of the steel slag provided a certain embedding depth for the asphalt mortar and formed a physical anchorage between the asphalt-binder and steel slag aggregates. This inherently contributed to improving the adhesive bonding and structural strength of the resultant asphalt mortar.

To accurately characterize the morphological properties of the aggregate surface texture, the SEM images were processed using MATLAB [78,79] to estimate the fractal dimensions of SEM grayscale images using the differential box counting algorithms [80] in the MATLAB subroutines. Table 3 lists the corresponding fractal dimensions of the aggregate surface texture, whilst Figure 9 presents the curved surface images formed using the gray value-coding of the SEM images.

**Table 3.** Fractal Dimensions of the aggregate Surface Texture.

| Aggregate Type | Steel Slag | Basalt |
| --- | --- | --- |
| Fractal dimension | 2.54 | 2.47 |

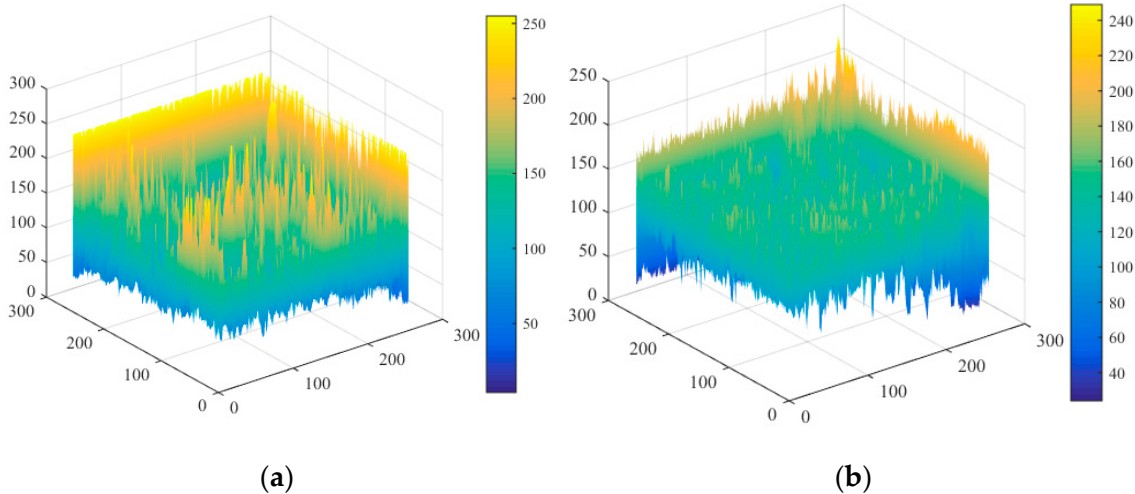

(a)                                                                (b)

**Figure 9.** Surface plot of SEM gray scale images: (**a**) Steel slag; (**b**) Basalt.

From Figures 7 and 8, it is evident that even the SEM images at ×200 magnification yielded a clear visual presentation of the aggregate surface texture, which partly ensured that the estimation of the aggregate texture structure had more statistical significance. Table 3, on the other hand, indicates that the fractal dimensions of the surface texture of the steel slag aggregates had increased by 3% than that of the basalt aggregates. This demonstrated that the surface texture of the steel slag aggregates was more outgrown and rougher than that of the basalt aggregates. This ultimately contributed to the stronger adhesion between the asphalt-binder and steel slag aggregates than that between the asphalt-binder and basalt aggregates under the same conditions. The corresponding SEM interfacial imaging results between the asphalt-binder and the aggregates are shown in Figures 10 and 11, respectively.

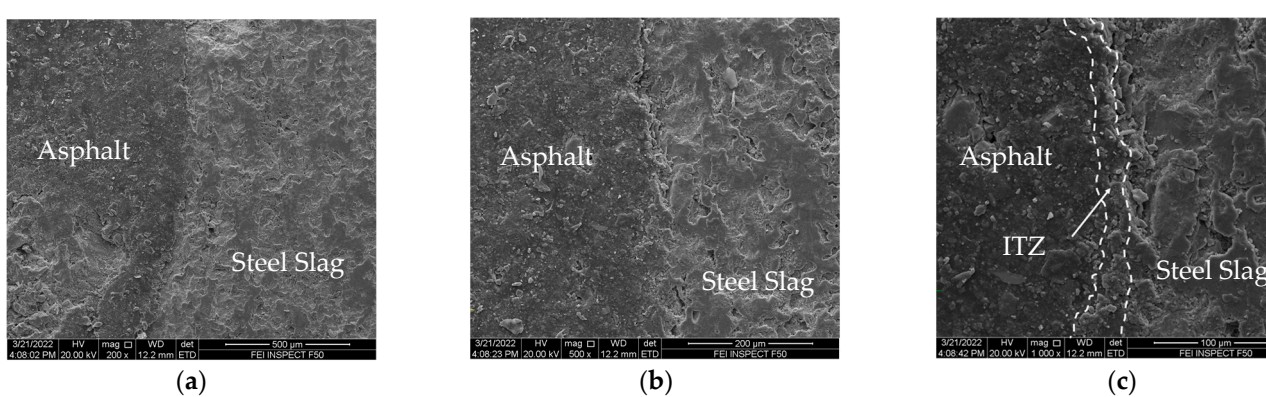

(a)                                    (b)                                    (c)

**Figure 10.** Asphalt-binder and steel slag interfacial imaging results: (**a**) ×200 magnification; (**b**) ×500 magnification; (**c**) ×1000 magnification.

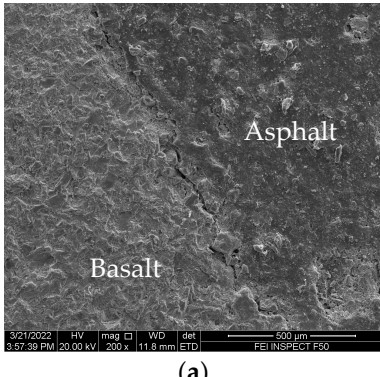 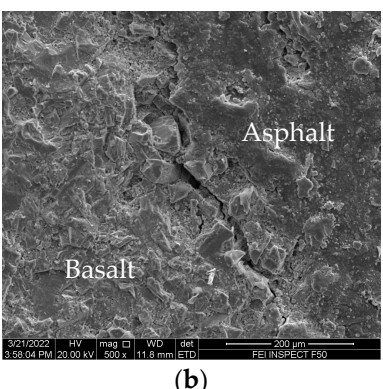 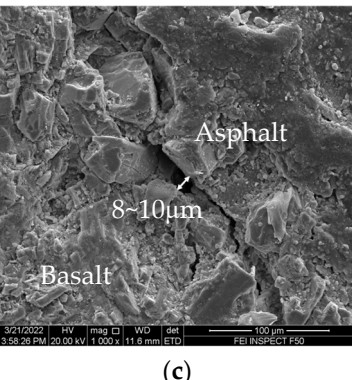

(a)  (b)  (c)

**Figure 11.** Asphalt-binder and basalt interfacial imaging results: (**a**) ×200 Magnification; (**b**) ×500 Magnification; (**c**) ×1000 Magnification.

Compared with Figure 11, Figure 10 visually shows that interface between the steel slag and asphalt-binder was continuous and uniform, whilst there were some gaps in the interface between the basalt and asphalt-binder (Figure 11). This is indicative that the interfacial bonding between the asphalt-binder and steel slag was better, with an equally strong adhesive contact between them. Because of the complex and rich pore texture structure on the surface of steel slag [81], the specific surface area of aggregates increased. This inherently made the asphalt-binder at the interface between the steel slag and the asphalt-binder to permeate through pores on the steel slag surface. Thereafter, the contact of the asphalt-binder and steel slag aggregates formed a certain embedding and anchoring depth mechanism to generate a composite phase between them. In agreement with the literature findings [77,82–84], it is apparent that the interfacial strength between the asphalt-binder and steel slag aggregates was enhanced due to the strong mechanical and chemical bonding force generated by the chemical reaction between the asphalt-binder and the steel slag. These strong bonds obviously aided in preventing the asphalt-binder from easily falling off (or stripping under water immersion) from the aggregates, which ultimately contributed to significantly enhancing the high-temperature rutting and moisture-damage resistance of the resultant asphalt mixture.

## 4. Conclusions

In this laboratory study, the production of asphalt mortar with steel slag and basalt aggregates (filler) at a fixed asphalt-binder-filler ratio of 0.4 was comparatively evaluated for its physical and rheological (DSR) properties including a quantitative assessment of the interaction between the asphalt-binder and aggregates (filler). Thereafter, the Fourier transform infrared (FTIR) spectrometer and scanning electron microscope (SEM) were then used to characterize the microscopic mechanism of the interaction and interfacial bonding between the asphalt-binder and the aggregates. Note we use "aggregates" to refer to fine-ground aggregates that pass through a 0.15 mm sieve and are retained on a 0.075 mm sieve, called, respectively, filler and aggregate-filler. From the study findings, the following conclusions and recommendations were drawn:

- The physical properties of the steel slag and basalt asphalt mortars were quantitatively similar and did not significantly differ from each other.
- The rheological properties of the steel slag asphalt mortar exhibited superiority over the basalt asphalt mortar, with the latter being more temperature sensitive and less rutting resistant. At intermediate temperatures, however, the difference in the rheological properties between the steel slag and basalt asphalt mortar was quantitatively insignificant.
- The parametric indices obtained from DSR rheological testing were found to be satisfactory for use as indicative measures to characterize and quantify the asphalt-binder-aggregate (filler) interaction capability. The corresponding results and findings indicated better interaction capability and interfacial bonding potential between the

asphalt-binder and steel slag aggregate-filler than that between the asphalt-binder and the basalt aggregate-filler.

- Based on the FTIR analysis, the main action between the asphalt-binder and basalt aggregates was predominantly physical. By contrast, the chemical bonding action between the asphalt-binder and steel slag aggregates that generated organic silicon compounds significantly contributed to enhancing the interfacial bond strength within the steel slag asphalt mortar.
- From SEM imaging analysis, the micro-surface texture of the steel slag aggregates was observed to be more overgrown and rougher than that of the basalt aggregates, which alluded to the improved adhesion between the asphalt-binder and steel slag aggregates. In addition to the chemical bonding force generated from the chemical reactions, there was also a strong mechanical bonding force that greatly enhanced the interfacial bond strength between the asphalt-binder and steel slag aggregate-fillers.

This laboratory study comparatively explored and provided a reference datum on the interfacial bonding mechanisms, physical properties, and rheological characteristics of steel slag aggregates as an admixture filler in the production of asphalt mortar. Whilst plausible results were obtained, recommendations for future follow-up studies should include colder-temperature testing for more in-depth cracking resistance evaluation and discrete moisture sensitive assessment of the asphalt mortars. Nonetheless, the study valuably adds to the state-of-the-art literature enrichment on the exploratory usage of steel slag and basalt aggregate-fillers to manufacture asphalt mortars with SBS-modified asphalt-binder.

**Author Contributions:** Conceptualization, X.C., X.Z. and W.W.; methodology, W.W. and Y.N.; validation, J.Z. and Z.L.; formal analysis, W.W. and Y.N.; investigation, Z.M.; resources, X.C. and Z.L.; writing—original draft preparation, W.W., Y.N. and Z.M.; writing—review and editing, X.C., X.Z., J.Z. and Z.L.; visualization, W.W.; supervision, X.C., X.Z. and J.Z.; funding acquisition, J.Z. and Z.L. All authors have read and agreed to the published version of the manuscript.

**Funding:** This research was funded by the National Key Research and Development Program (Grant No. 2018YFB1600200), the National Science Foundation of China (Grant No. 51778140 and No. 52078130), Suzhou Jiaotou Construction Management Co., Ltd. (Grant No. 8521008638), and Suzhou Sanchuang Pavement Engineering Co., Ltd. (Grant No. 8521008758).

**Institutional Review Board Statement:** Not applicable.

**Informed Consent Statement:** Not applicable.

**Data Availability Statement:** Not applicable.

**Acknowledgments:** Special thanks and due gratitude go to all those who offered the administrative and technical support during the course of this study.

**Conflicts of Interest:** The authors have received research grants from Suzhou Jiaotou Construction Management Co., Ltd. and Suzhou Sanchuang Pavement Engineering Co., Ltd. Author J.Z. is employed by Suzhou Jiaotou Construction Management Co., Ltd., and author Z.L. is employed by Suzhou Sanchuang Pavement Engineering Co., Ltd.

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
