# Peer review of "Research on the Interaction Capability and Microscopic Interfacial Mechanism between Asphalt-Binder and Steel Slag Aggregate-Filler"

_coatings, doi:10.3390/coatings12121871_

Round 1
Reviewer 1 Report
The authors reported possible interactions between asphalt-binder and steel slag aggregate-filler; the manuscript is well organized and documented, the expected outcome was clearly achieved. However, the reason why these interactions are important was not clear enough. The authors could insert a justification so that the aim of the study to be clearer.
Author Response
Thank you very much for your comments and professional advice.
Please see the attachment.

Reviewer 2 Report
Dear authors,
The manuscript focused on the Interaction Capability and Microscopic Interfacial Mechanism between Asphalt-Binder and Steel Slag Aggregate-Filler, which is organized well; however, it needs further improvement to match with the Journal criteria. Please see the below comments:
1) Many references were cited in the paper; however, most are older than 2017. The authors encouraged to use of the most recent relevant research, especially through the introduction.
2) Any special reason to use a local SBS, Basalt and steel slag provider which may limit the result achieved for this research?
3) The Asphalt Mortar preparation is most like a lab scale; however, there would be a slight difference between the practical onsite preparation. Can authors add any influence which may affect the further results?
4) The Physical Property Testing were referenced to another work, while the author should place all the necessary information within the same section. It makes the reader confused among the relevant sections.
5) Figure 4a shows a significant difference between Parametric Index Results; however, no clear explanation is found in the relevant paragraph.
6) The microstructural differences were well explained; however it did not go deeply into the form of structures or any phase differences. an EBSD analysis may help to understand it more in-depth.
7) English need to be checked by a technical native English speaker.
Author Response

(The authors gave the same response as above.)

Reviewer 3 Report
The research on the interaction capability and microscopic interfacial mechanism between asphalt-binder and steel slag aggregate is interesting. But…bituminous binder mixed with a mineral filler leads to mastic asphalt. Therefore, instead of asphalt mortar, the phrase mastic asphalt should be used. When obtaining mastic asphalt, a filler with a grain size of ≤0.075 mm should be used. For such a mixture, it is correct to use the tests used for testing bitumen binders. The authors used aggregate with a grain size of 0.075 mm to 0.15 mm in the work. Therefore, the application of Penetration, softening point and ductility tests may be questionable.
Author Response

(The authors gave the same response as above.)

Round 2
Reviewer 3 Report
Thank you for responding to the review.